# Nanobiotechnological Approaches to Enhance Drought Tolerance in *Catharanthus roseus* Plants Using Salicylic Acid in Bulk and Nanoform

**DOI:** 10.3390/molecules27165112

**Published:** 2022-08-11

**Authors:** Dina Salem, Hoda A. S. El-Garhy, Ismail A. Ismail, Eldessoky S. Dessoky, Bassem N. Samra, Tahsin Shoala

**Affiliations:** 1Agricultural Biotechnology Department, College of Biotechnology, Misr University for Science and Technology, Giza 12563, Egypt; 2Genetics and Genetic Engineering Department, Faculty of Agriculture, Benha University, Qalyubia 13736, Egypt; 3Department of biology, College of Science, Taif University, P.O. Box 11099, Taif 21944, Saudi Arabia; 4Environmental Biotechnology Department, College of Biotechnology, Misr University for Science and Technology, Giza 12563, Egypt

**Keywords:** drought stress, salicylic acid nanoparticles, drought tolerance genes, alkaloids, growth parameter, qRT-PCR

## Abstract

Drought has a detrimental effect on crop production, affecting economically important plants’ growth rates and development. *Catharanthus roseus* is an important medicinal plant that produces many pharmacologically active compounds, some of which have significant antitumor activity. The effect of bulk salicylic acid (SA) and salicylic acid nanoparticles (SA-NPs) were evaluated on water-stressed *Catharanthus roseus* plants. The results showed that SA and SA-NPs alleviated the negative effects of drought in the treated plants by increasing their shoot and root weights, relative water content, leaf area index, chlorophyll content, and total alkaloids percentage. From the results, a low concentration (0.05 mM) of SA-NPs exerted positive effects on the treated plants, while the best results of the bulk SA were recorded after using the highest concentration (0.1 mM). Both treatments increased the expression level of *WRKY1*, *WRKY2*, *WRKY40*, *LEA*, and *MYC2* genes, while the mRNA level of MPKK1 and MPK6 did not show a significant change. This study discussed the importance of SA-NPs in the induction of drought stress tolerance even when used in low concentrations, in contrast to bulk SA, which exerts significant results only at higher concentrations.

## 1. Introduction

*Catharanthus roseus* is an evergreen medicinal plant, belonging to the family Apocynaceae, containing many medicinal compounds. It has a long history of being used to treat a variety of diseases, including diabetes [1] and gonorrhea [2]. Vincristine and vinblastine are two alkaloids derived from the stem and leaves of this plant; they have been used to treat different types of cancer in combination with another chemotherapeutic. *C. roseus* contains about one hundred and twenty alkaloids, seventy of which have pharmacological importance [3]. The production of these secondary metabolites is widely affected by different biotic and abiotic stress conditions [4]. 

Abiotic stresses, including salinity, drought, heavy metals, and flooding, are major factors hindering crop productivity around the world. The previous investigations reported that 90% of the agricultural lands are vulnerable to these abovementioned stress factors [5]. Drought and heat are the most dangerous abiotic stress factors, impairing crop production in large areas around the world. The negative effect of droughts on plant growth is attributed to the decrease in water-soluble nutrients, which inhibits the photosynthetic process by damaging the photosynthetic pigments, thus decreasing chlorophyll and thereby reducing crop productivity [6,7]. Drought has also slowed plant growth by interfering with cell elongation and mitosis [8]. The harmful effect of moisture stress on yield is attributed to various factors such as a decreased rate of photosynthesis [9], disturbed assimilate partitioning via an assimilation of the migration to the roots to increase water uptake [10], or poor leaf development [11]. 

The plant’s response to drought varies depending on the season, plant species, the plant growth stage, and the stress severity [12]. The plant’s first response to a change in water potential is stomatal closure, which results in a decrease in carbon dioxide and oxidative damage; thus, no assimilation takes place [13]. Water stress can cause physiological and morphological changes in *C. roseus*. A foliar application of SA and L-arginine reduced the negative effects of water stress by improving the morpho-physiological characteristics of C. roseus plants under drought stress. Under 70 percent FC stress, L-arginine at 3 mM and SA at 100 mg/L increased the nitrogen and phosphorus content as well as the number and diameter of flowers, total chlorophyll content, and root and shoot growth in *C. roseus* plants [14]. Many studies have documented the negative effects of drought on *C. roseus*. Drought has been shown to negatively affect plant growth parameters such as plant weight, height, and chlorophyll content [15], as well as the accumulation of stress-related salts such as proline and glycine betaine. Several plant secondary metabolites were also increased to assist *C. roseus* in adapting to the negative effects of drought [16]. Plant tolerance to abiotic stress conditions is an important feature of many plants, particularly economic crops. In this regard, various strategies have been used, including genetic engineering approaches, plant breeding, and the use of growth regulators.

Salicylic acid (SA) is a phenolic phytohormone with an important role in plant growth, development, and stress tolerance [17]. It can promote photosynthesis, CO_2_ assimilation, ribulose-1,5-bisphosphate carboxylase/oxygenase activity, stomatal conductance, and increase chlorophyll and carotenoid content [18]. The application of SA in plants under stress conditions promotes plant defenses through increasing the expression of genes coding for antioxidant enzymes, consequently inducing stress-responsive genes [19,20]. Importantly, SA could regulate the transcription of 100 s of defense genes [21,22,23]; in addition, it promotes proline accumulation [24]. 

Nanotechnology has attracted great interest in the last few years because of its increased number of applications. Nanoparticles have more solubility, surface area, and reactivity compared to bulk material [25]. Therefore, they have gained a promising position to ameliorate the harmful effects of abiotic and biotic stress to achieve the goal of sustainable agriculture globally [26]. Due to their impact on abiotic stress tolerance and the nutritional quality of crops, the research related to the application of nanoparticles is increasing. Several types of nanoparticles have been tested for their protective effect against biotic and abiotic stresses [20,26,27,28,29]. 

Multiple WRKY genes contribute to the drought stress response through distinct signaling pathways, making it one of the largest transcription factor families among higher plants [30,31]. WRKYs can play a role in the response to hormone signals. Salicylic acid (SA) can boost the production of the WRKY70 transcription factor, demonstrating how WRKYs can be involved in the response to hormone signals [32]. WRKY genes play a critical role in plant growth and development, abiotic stress tolerance, and other biological processes by controlling the expression of target genes [33,34]. MAPKs target other kinases, enzymes, or transcription factors in the cytoplasm or nucleus, thereby controlling cell signaling in plants and thus their ability to adapt to environmental stress [35]. Late Embryogenesis-Abundant Proteins (LEA) are enhancing the activities of antioxidant enzymes and act as chaperone proteins to protect cells against membrane damage in drought-stressed plants [36,37,38]. MYC transcription factors have a role in the regulation of crosstalk between the signaling pathways of JA and those of other phytohormones such as abscisic acid (ABA), SA, gibberellins (GAs), and auxin, which have a role in drought tolerance through regulating stomatal closure [39].

The goal of this research is to use various SA concentrations in bulk and nano form to reduce the effects of drought stress on *Catharanthus roseus* plants. It also evaluates the effects of these concentrations on morphological, physiological, and biochemical traits, as well as stress-responsive gene expression in stressed plants.

## 2. Materials and Methods

### 2.1. Preparation of Salicylic Acid Nanoparticles

Salicylic acid (SA) was purchased from Sigma-Aldrich (CAS number: S5922, BioXtra, ≥99.0%). A total of 20 mg of SA was weighed and dissolved in 20 mL of absolute ethanol and sonicated with an ultrasonic power (XUBA3 Analogue Ultrasonic Bath, Grant Company, Devizes, UK) and a frequency of 50 kHz for an hour at room temperature (25 °C) [40].

#### 2.1.1. Characterization of Nanomaterials Using Dynamic Light Scattering (DLS) 

The distribution and particle size of SA-NPs were determined using a Zetasizer Nano ZS and a dynamic light scattering approach (Malvern Instruments, Malvern, UK). A total of 30 µL of the generated nanoparticles were diluted in 3 mL of deionized water at room temperature before testing. The particle size was estimated by taking the mean of the Z-averages of three distinct batches of each of the aforementioned nanoparticle types.

#### 2.1.2. Transmission Electron Microscopy (TEM) 

Transmission Electron microscopy was utilized to study the morphology of all three kinds of nanoparticles. In this case, a drop of the nanoparticles solution was sonicated before being placed on carbon-coated copper grids (CCG) and allowed to dry completely by allowing water to evaporate at ambient temperature. Electron micrographs were taken at Al-Azhar University’s Regional Center for Mycology and Biotechnology (RCMB) using a JEOL GEM-1010 transmission electron microscope set at 70 kV [41].

### 2.2. Plant materials, Drought, and SA Treatments

The seeds were sterilized by soaking them for 3 min in 70% percent Ethanol and then rinsing them several times with sterile H_2_O. *Catharanthus roseus* (Cape periwinkle) seeds were sterilized and sown in plastic pots filled with a 5:1 (*v*/*v*) mixture of clay soil and peat. Treatments were administered between 45 and 60 days. Pots were exposed to drought stress alone while maintaining 25% field capacity, and other pots were exposed to drought stress as well as different concentrations of salicylic acid nanoparticles (0.05 mM and 0.1 mM) and bulk salicylic acid (0.05 mM and 0.1 mM) by foliar application, while other pots were irrigated to keep the soil at the appropriate moisture level or left untreated as a control, or treated with the mentioned concentrations of bulk salicylic acid and salicylic acid nanoparticles. Five pots were used for each treatment, and plants were treated with SA and SA-NPs three times with interval of five days. 

### 2.3. Measuring Growth Parameters

The growth parameters of treated and untreated 60-day-old *Catharanthus roseus* plants were measured using five plants of each treatment, and the fresh weights of shoots and roots were recorded first. Then, plants were left to dry at 60 °C for 72 h to measure the dry weight of the shoots and roots and the root–shoot ratio. In addition, the leaf area index (LAI) was determined using the following formula as suggested by Watson (1947) [42].
LAI = Leaf area plant/Area occupied the plant(1)

### 2.4. Relative Water Content (RWC) Measurement

Leaves of treated and untreated 60-day-old plants were cut and sealed in plastic bags and their fresh weights were recorded immediately after cutting.. Then, they were soaked in distilled water for twenty-four hours at room temperature to record the turgid weight. Finally, turgid leaves were dried in an oven for 72 h at 70 °C to record their dry weight. RWC was calculated using the formula of Schonfeld et al. (1988) [43]:RWC (%) = (FW − DW)/(TW − DW) × 100(2)
where FW = Fresh weight, DW = Dry weight, and TW = Turgid weight

### 2.5. Chlorophyll a, b and Carotenoid Content Measurement

Total chlorophyll a and b and carotenoid contents in fresh leaves from 60-day-old plants were measured using the method of Arnon (1949). One gram of fresh tissue of leaves was ground in a mortar and pestle using 10 mL of 80% acetone. The optical densities (OD) of the solutions obtained from different treatments were recorded at 663, 645, 480, and 510 nm using a spectrophotometer. The content of the three photosynthetic pigments was measured using the following formulas and values, which were expressed in mg g-1 FW [44]: Chlorophyll-a = [12.7 (OD663) − 2.69 (OD645)] × V/1000 × W(3)
Chlorophyll-b = [22.9 (OD645) − 4.68 (OD663)] × V/1000 × W(4)
Carotenoid = [7.6 (OD480) − 1.49 (OD510)] × V/1000 × W(5)
where V = volume of the extract (mL) and W = weight of the fresh leaf (g).

### 2.6. Total Alkaloid Measurement

Total Alkaloid content was measured using Harborne’s (1973) method. Briefly, five grams of air-dried leaves was weighed and then ground to powder. Then, it was incubated for four hours in 250 mL of 10% acetic acid in ethanol. A water bath was used to evaporate and concentrate the solution until it was one-fourth the original volume of the extract after the solution had been filtered to remove debris. Then, ammonium hydroxide was added dropwise to the concentrated extract until the precipitation process was complete and the obtained extract was allowed to settle. This step was followed by the collection and washing of precipitate using diluted ammonium hydroxide. Finally, the precipitate was filtered, dried, and weighed, and the percentage of total alkaloids was determined by dividing the obtained weight by the weight of the used air-dried leaves [45]. 

### 2.7. RNA Extraction and Gene Expression Profiling Using Quantitative Real-Time PCR (qRT-PCR)

The total RNA was extracted and purified using a total RNA extraction kit (Bioflux, Beijing, China) from the treated and untreated *Catharanthus roseus* leaves. qRT-PCR was performed with ABT One-Step SYBR kit (Rox) with an RT-PCR Kit, using 100 ng total RNA/reaction, RNasin Ribonuclease Inhibitor (Promega Corporation, Madison, WI, USA), and gene-specific primers (Table 1). Triplicate PCR reactions, as well as RNA template negative and non-template controls, were carried out and used. Each reaction contained 2.5 μL of RNA (except for NTC), 10 μL of 2× reaction buffer, 0.8 μL of superscript III-RT platinum Taq mix, 0.5 μL of SYBR green dye, and 0.3 mM of each forward and reverse primer of the seven studied genes (Table 1). The final volume was adjusted to 20 μL by adding RNase-free water. Reactions were measured using Prime Q real-time qPCR equipment (Techne, London, UK) with a one-step cycling protocol at 45 °C for 10 min, 95 °C for 10 min, 45 cycles of 95 °C for 15 s, and then 60 °C for 30 s followed by 72 °C for 30 s. For qPCR data normalization, the *actin* gene (GenBank MG813871.1) was used as an internal reference gene. All experimentally generated variations in the expression of the genes under investigation are represented as n-fold changes in comparison to the controls. The formula RQ = 2^–ΔΔCT^ was used to calculate the relative gene expression ratios (RQ) between the treatment and control groups [46].

### 2.8. Statistical Analysis

Analysis of variance (ANOVA) was carried out using SPSS version 19 (SPSS Inc., Chicago, IL, USA). LSD test was conducted to test the significance between mean values (*p* < 0.05).

## 3. Results 

### 3.1. Characterization of SA Nanoparticles

The prepared SA nanoparticles were characterized by different methods to analyze their particle sizes and aggregation states. A dynamic light scattering technique was performed to understand the stability and the size distribution of the prepared SA-NPs. A TEM (transmission electron microscope) analysis showed that the SA-NPs were widely dispersed and composed of particles with an average of <100 nm. Moreover, the TEM micrograph showed that the SA-NPs had a spherical morphology with an average size between 5.71 and 17.3 nm (Figure 1a). The zeta potential results for the SANPs showed a positive surface charge with a value of approximately 3.99 mV (Figure 1b). The analysis showed that the size distribution range was within 20 nm and the curve showed a monomodal distribution (polydispersity index PDI), as shown in (Figure 1c).

### 3.2. Plant Growth Response

Drought stress (25% field capacity) significantly (*p* < 0.05) affected *Catharanthus roseus*’s growth parameters in terms of plant height, shoot and root dry weight, and leaf area index. The results showed that all these parameters had improved by spraying the plants with all the studied concentrations of SA and SANPs under drought and normal irrigation conditions (Table 2). Under drought stress, both treatments increased the plant height and plant weight compared to the control. Plants treated with 0.1 mM of the SANPs recorded the highest plant height (55 cm), a 14.58% increase, compared to the untreated plants under drought stress. Whereas the highest shoot and root dry weights (128% increase) were evidenced in the plants treated with 0.05 mM of the SANPs under drought stress (2.4 and 3.35 gm, respectively) compared to the untreated stressed plants. The root–shoot ratio also decreased significantly (*p* < 0.05) in response to drought stress, and this harmful effect was adverted by all the studied treatments.

### 3.3. Chlorophyll a and b and Carotenoid Content 

Drought stress significantly (*p* < 0.05) decreased chlorophyll a (Chl-a) content (0.75 mg/gm) compared to the control. However, using SA in both types of formulations increased the chl-a and chl-b content; the highest chlorophyll content was recorded in the plants treated with 0.05 mM of SANP at chl-a (0.96 mg/gm), a 25.33% increase, compared to the untreated plants in underwater deficiency conditions. Whereas the highest increase in carotenoid content (0.31 mg/gm) was recorded in drought-stressed plants treated with 0.05 mM of SA, which increased carotenoids 16% compared to the control (Table 3). 

### 3.4. Relative Water Content and Leaf Area Index

Water deficiency significantly (*p* < 0.05) reduced the relative water content (RWC) in the plant leaves (Table 4). Under drought conditions, spraying plants with 0.05 and 0.1 mM of SA improved the RWC by 71.66% and 85.33%, respectively, while using SANPs (0.05 and 0.1 mM) improved the RWC value by 86.1% and 89.17%, respectively. The leaf area index was also negatively affected by drought stress: water deficiency decreased the leaf area in the untreated plants by 20.5% compared to the normally irrigated plants. On the other hand, the two used treatments exerted a significant effect on alleviating this inhibition. The highest increase in the leaf area index was evidenced in plants under drought stress treated with 0.05 mM of SANP (143% increase), followed by 0.1 mM of the SANP treatment (4.87), a 135.27% increase compared to the control. 

### 3.5. Alkaloid Content

The results in Table 5 show that the drought stress increased the alkaloid content (0.77) compared with its content in the normally irrigated control (0.72). The highest increase in alkaloid content was recorded in the plants treated with 0.05 mM SANP, which increased alkaloid content by 0.83 followed by 0.82 in the plants treated with 0.1 mM SA compared to the untreated plants under drought stress. 

### 3.6. Effect of Different Treatments on Expression Profiling of Drought Tolerance Associated Genes Using qRT-PCR

The expression levels of seven drought tolerance-associated genes—LEA, WRKY1, WRKY2, WRKY40, MYC2, MPK1, and MPK6—were upregulated under elicitation by the SA and SANPs at 0.05 and 0.1 mM. The results showed that the mRNA level of the LEA gene increased in the plants treated with both SA and SANPs; the highest expression increase was quantified in the plants treated with 0.05 mM SA—a 5.6- fold increase (Figure 2). In Figure 3A,B, WRKY1 and WRKY2 expression levels increased in response to SA and SANPs at both concentrations, except for the expression of the WRKY40 gene (Figure 3C), which was highly upregulated under the effect of foliar application with 0.05 and 0.1 mM SANPs, (4.3 and 4.6- fold increase, respectively), compared with other treatments as well as the negative control, untreated plants under drought stress. The mRNA level of the MYC2 transcription factor gene increased by different concentrations of both forms of salicylic acid, and the highest increase was recorded in the plant leaf samples of the plants treated with 0.1 mM SANP (5.1- fold increase) followed by 0.05 mM (3.8- fold increase), as shown in Figure 4. Whereas the MPKK1 and MPK6 genes showed a slight upregulation in the plants treated with SA and SANPs at both the studied concentrations compared with their expressions in untreated plants under drought stress—the negative control (Figure 5).

## 4. Discussion

Drought significantly reduced the fresh and dry weights of shoots and roots, as well as their length, according to the findings. Drought, similar to other abiotic stressors, causes cellular oxidative stress and reactive oxygen species production [47]. In low and moderate concentrations, these free radicals act as second messengers, activating plant tolerance. However, at high concentrations, reactive oxygen species are destructive to DNA, proteins, and biological membranes, and they also inhibit enzyme activity, including photosynthesis enzymes. On the other hand, the water-stressed plants treated with bulk SA acid and SA-NPs mitigated the negative effects of drought on growth parameters. These results are in line with the previous studies performed on different plants including *Zea mays* [48], *Conocarpus erectus* L. and *Populus* sp. [49], and *Rosmarinus officinalis* L. [50].

In contrast to bulk SA, SA-NPs have a mitigating effect at low concentrations; however, their results were only significant at higher concentrations (0.1 mM). Salicylic acid promotes plant growth by increasing stomatal conductance and improving water relations, which promotes photosynthesis and increases plant growth [51]. Furthermore, by inducing root formation, nutrient uptake, activating photosynthetic enzymes, and cell division, SA helps plants mitigate the negative effects of moisture deficiency on growth.

Water deficiency decreased chlorophyll content in this study; the decrease in the photosynthetic pigments in the stressed plants had previously been demonstrated by Munne-Bosch and Alegre (2000) [52] in different species. This effect was caused by the action of released reactive oxygen species on the thylakoid membrane; chlorophyll degradation by chlorophyllase, which is activated in drought-exposed plants; and the decreased activity of chlorophyll synthase [53,54]. In water-stressed plants, salicylic acid reversed this effect, and the chlorophyll content increased significantly. These findings are attributed to salicylic acid’s protective role in preserving plant membranes, including the thylakoid membrane, by activating antioxidant enzymes and non-enzymatic antioxidants.

Furthermore, salicylic acid activates photosynthetic enzymes while inhibiting chlorophyll oxidase, allowing chlorophyll degradation to continue. Our findings are consistent with those of Ghazali and Estaji (2020), who conducted research on *Capsicum annuum* [55].

Many plants showed a decrease in the leaf area index in response to water deficiency [56,57,58,59]. The leaf area is affected by assimilates and turgor pressure, both of which reduce water stress. In the presence of moisture stress, various plants tend to reduce the area of young leaves and the expansion of developed leaves to reduce transpiration. Salicylic acid, on the other hand, increased leaf area index in both water-stressed and well-watered plants. Salicylic acid increased the leaf area index by increasing the turgor pressure via assimilates, proline, and soluble proteins’ accumulation [50].

The results revealed a significant decrease in the relative water content in plants subjected to drought stress without the application of salicylic acid. These findings are consistent with the findings of Khazei and Estaji (2020) [55], who discovered that drought significantly reduced the relative water content. This reduction could be due to an increased membrane permeability and a decreased water uptake. RWC is a good indicator of drought resistance; in other words, plants with a high RWC should have higher yields under drought stress [60].

In all treatments, the total percentage of alkaloids increased when compared to the control, with the highest recorded value obtained when treating plants with the highest concentration of salicylic acid nanoparticles with drought stress (0.82 percent) when compared to untreated plants under drought stress (0.77 percent). Similar findings were reported by Ababaf et al. (2021) [61], who investigated the effect of salicylic acid and a combination of salicylic acid and jasmonic acid on a drought-stressed *C. roseus* plant. Under drought conditions, the expression of alkaloid biosynthesis enzymes such as strictosidine synthase, tryptophan decarboxylase, and deacetylvindoline-4-Oacetyltransferase increases, resulting in an increase in alkaloids.

SA increased the synthesis of a variety of secondary metabolites in stressed plants, including alkaloids, phenols, terpenes, phenolics, glucosinolates, phytoalexins, thionins, defensins, and allicin [62] by eliciting genes responsible for their production [63].

In addition to inducing stomatal closure to prevent further water loss in plants under drought stress, salicylic acid treatment induces the production of secondary metabolites that scavenge reactive oxygen species to protect the plants from lipid peroxidation and other oxidative damage-related harmful effects [64]. Secondary metabolite production is modulated by various transcription factors such as WRKY, bHLH, and MYB, which are induced by stress conditions, plant hormones such as salicylic acid, MAPK, and reactive oxygen species to ultimately affect the expression of secondary metabolites related to genes [65].

Plant gene expression and associated biological pathways can be altered by nanoparticles, affecting plant growth and development [66]. The results showed that the foliar application of SA at 0.05- and 0.1-mM increased LEA gene expression in plants under drought stress. The induction of the dehydrin gene, one of the LEA genes, in response to salicylic acid was previously observed in drought-stressed barley plants treated with 0.2 mM salicylic acid [67]. LEA is a plant protein class that accumulates in response to environmental stress and serves specific protective functions in plant cells. Dehydrins have been shown in vitro to be capable of membrane stabilization, protein aggregation prevention, cryoprotection, nucleic acid protection, ROS scavenging, and the binding of small ligands such as water, ice crystals, and metal ions [68].

Multiple WRKY genes contribute to the drought stress response via distinct signaling pathways, making these genes one of higher plants’ largest transcription factor families [28,29]. The current study found that SA and SA-NPs induced the expression of WRKY1 and WRKY2 genes in plants under drought stress at both concentrations. While a field application of 0.05 and 0.1 mM SANPs on drought-stressed plants increased WRKY40 mRNA levels compared to negative control plants, untreated drought-stressed plants did not experience the same increase. According to Gao et al. (2018), WRKY2 plays an important role in the stress response, and the TaWRKY2 promoter contains cis-elements involved in plants’ signaling of stress-related hormones such as methyl jasmonate acid (MeJA) and salicylic acid (SA) [69].

SA can both stimulate and inhibit the expression levels of WRKY70 transcription factor [30]. Furthermore, it regulates target gene expression by identifying and binding W-box cis-elements or interacting with other regulatory factors, and thus plays an important role in plant growth and development, abiotic stress tolerance, and other biological processes [32,34,70]. Pathogens or salicylic acid were found to induce WRKY gene expression in Arabidopsis [71]. The overexpression of ZmWRKY40 in *Zea mays* reduced the ROS content and increased the activities of POD and CAT under drought conditions, as well as activating the expression of the stress-responsive genes STZ, DREB2B, and RD29A [72].

In general, WRKYs mediate the salicylic acid response to stress-enhancing growth parameters, with plants overexpressing these transcription factors showing an increase in their roots’ dry weight [73].

Furthermore, different treatments increased the mRNA level of the MYC2 transcription factor gene, with the highest fold change recorded in the plants treated with 0.1 mM of SA-NPs (5.1-fold increase). When compared to the negative control, the same concentration of bulk salicylic increased it by 2.4-fold. MYC2 is an ABA-related gene that belongs to the bHLH transcription factor family, which confers stress tolerance in plants. Salicylic acid stimulates H_2_O_2_ accumulation in the early stages of drought (0–6 days), followed by H_2_O_2_-stimulated ABA accumulation and the induction of ABA signaling and responses [74].

In comparison to the negative control, the foliar application of SA and SA-NPs at 0.05 and 0.1 mM slightly increased the expression of MPK1 and MPK6 genes in the plants under drought stress. MAPK transcription factors play critical roles in the regulation of stress-related hormone signaling, the activation of stress-related responses, tissue differentiation, and metabolism [75]. Furthermore, it has been demonstrated that the upregulation of MAPKs induces stress tolerance [76].

The results showed that the SA-NPs exerted significant effects even at very low concentrations when compared to the bulk material due to their greater solubility, surface area, reactivity sorption capacity, and controlled-release kinetics to targeted sites, making them a “smart delivery system.” All of these properties increase input efficiency, reduce relevant losses, and reduce environmental pollution [26], especially when the nanoparticles used are produced naturally in plants, such as salicylic acid. Nanoparticles stimulate proline production and protect membranes from damage. Several studies have also shown that nanoparticles can activate stress-related genes (Figure 6) [77].

To deal with both biotic and abiotic stress, nanoparticles may stimulate the production of secondary metabolites. They enter plant cells, causing membrane damage and the production of reactive oxygen species, which activates antioxidant production and mitogen-activated protein kinase, inducing secondary metabolite production genes (Figure 6) [78].

## 5. Conclusions

The foliar spraying of in *C. roseus* with bulk SA and SA-NPs was found to be very effective in reversing the negative effects of water stress. The foliar application of SA-NPs, especially at 0.05 mM, improved growth, photosynthetic pigment biosynthesis, and cellular osmotic adjustment. Water stress tolerance was also associated with antioxidative defense mechanisms, processes such as nutrient uptake enhancement, the induction of root formation, and increased cell division in the apical region of the root meristem; ultimately, it results in an increase in plant growth and an increase in secondary metabolites such as alkaloids to deal with stress conditions. Furthermore, SA increases the expression of transcription factors and genes involved in stress tolerance. Therefore, foliar sprays containing bulk SA and SA-NPs were very effective in mitigating the negative effects of water stress in *C. roseus*.

## Figures and Tables

**Figure 1 molecules-27-05112-f001:**
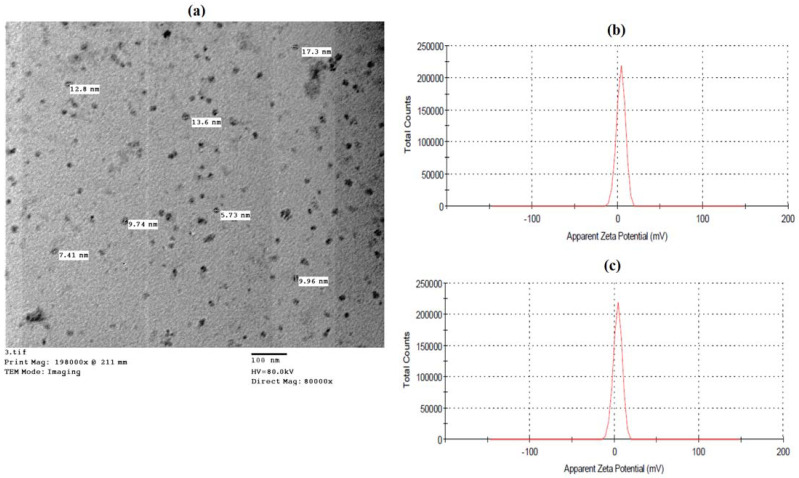
Characterization of salicylic acid nanoparticles using Transmission electron microscopy (TEM), Zeta potential, and Zeta sizer. (**a**) Transmission electron microscopy (TEM) of SA-NPs, (**b**) zeta potential of SA-NPs, and (**c**) average size of SA-NPs.

**Figure 2 molecules-27-05112-f002:**
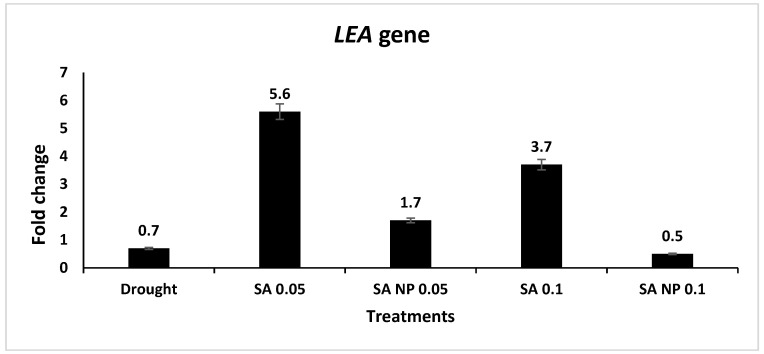
Expression profiling of *LEA* gene expression in *Catharanthus roseus* leaves of plants treated with SANPs and bulk SA at 0.05 and 0.1 mM, using qRT-PCR analysis. The *actin* gene (MG813871.1) was used as an internal reference gene for data normalization. Error bars represent standard errors.

**Figure 3 molecules-27-05112-f003:**
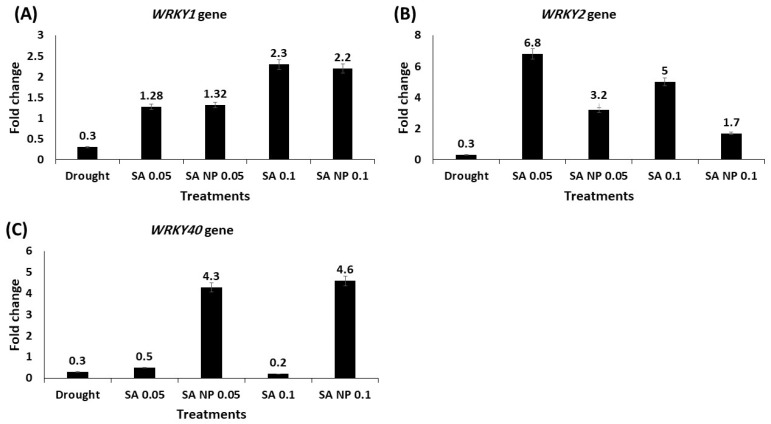
The effect of SANPs and bulk SA at 0.05 and 0.1 mM as a foliar application on the expression profiling of drought-tolerance related genes in *Catharanthus roseus* leaves of treated plants using qRT-PCR analysis. (**A**) *WRKY1* gene; (**B**) *WRKY2* gene; (**C**) *WRKY40* gene. Data were normalized using the *Actin* gene (MG813871.1) as an internal reference gene. Error bars represent standard errors.

**Figure 4 molecules-27-05112-f004:**
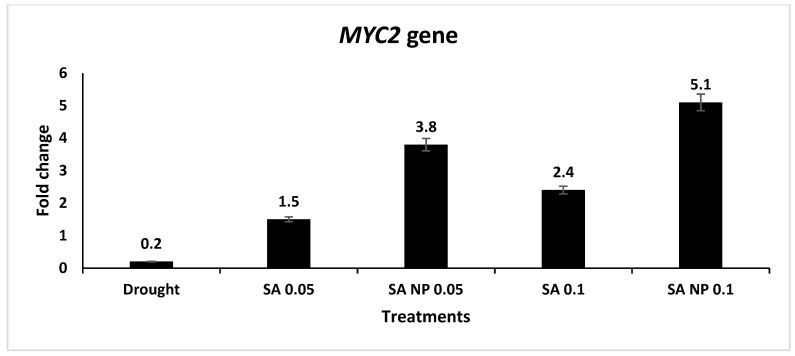
Expression profiling of *MYC2* gene expression in *Catharanthus roseus* leaves of plants treated with SANPs and bulk SA at 0.05 and 0.1 mM, using qRT-PCR analysis. The *actin* gene (MG813871.1) was used as an internal reference gene for data normalization. Error bars represent standard errors.

**Figure 5 molecules-27-05112-f005:**
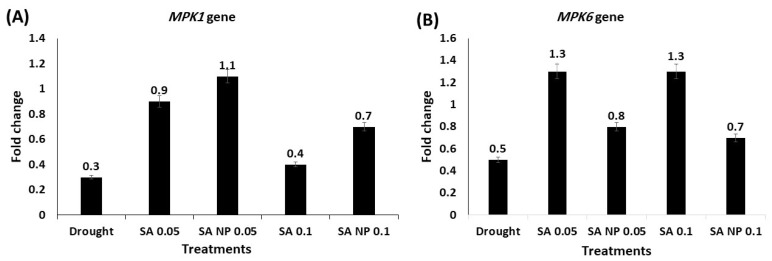
The effect of SANPs and bulk SA at 0.05 and 0.1 mM as a foliar application on the expression profiling of drought-tolerance related genes in *Catharanthus roseus* leaves of treated plants using qRT-PCR analysis. (**A**) *MPK1* gene; (**B**) *MPK6* gene. Data were normalized using the *Actin* gene (MG813871.1) as an internal reference gene. Error bars represent standard errors.

**Figure 6 molecules-27-05112-f006:**
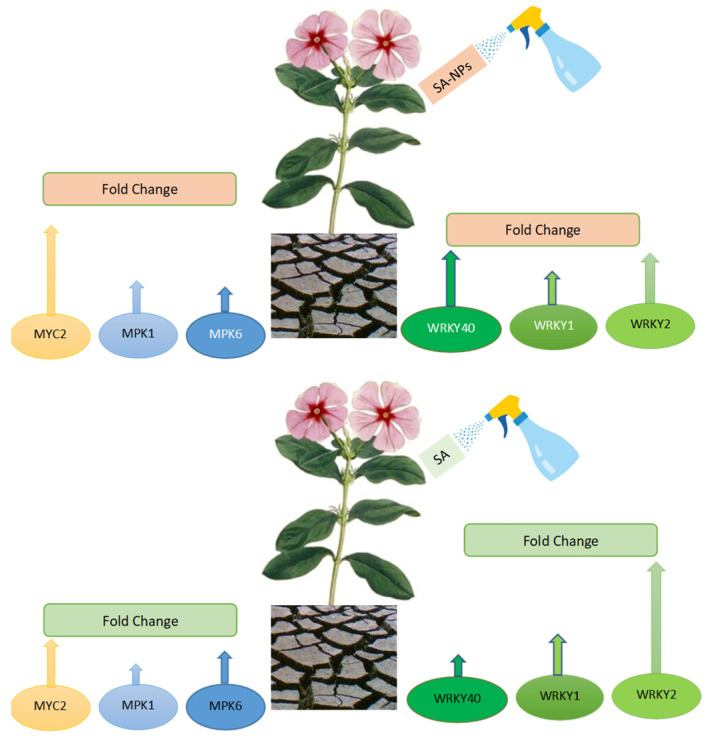
Schematic diagram of drought stress, exogenous application with SA and SANPs, and genetic response for *Catharanthus roseus* plants.

**Table 1 molecules-27-05112-t001:** Gene IDs and primer sequences for the selected genes used for qRT-PCR.

Accession Number	Gene Name	Forward Primer	Reverse Primer
HQ646368.1	*WRKY1*	ACCACGATGGCTTGGCATGA	TGTACGGAGAGTTCGGCAGC
JX241693.1	*WRKY2*	TGCAGGCTCAAAATGGCAGC	CACTGCCATGTGTTGCACGA
MG676674.1	*WRKY40*	TCAGGAGGAGGACCAACAATTAC	TGAAATGGCGGCTGCAAGTG
KR703580.1	*MPK6*	TCCTCCGTCTCAGCAACAGC	ATGGCTAAGAACGGCCGGAA
KR703579.1	*MPKK1*	CTATCTCCTCCGCCGCTACG	GCCGTTACCGTGGCCTAAGA
AF283507.2	*MYC2*	ATTTCTTTGTGGCCGCCGTC	AAGCAGGTTGGGCAGGCATA
DQ016338.1	*LEA*	AAGAGACAGCAGAGGCAGGG	TCAGCAGCACCTTGAGCCAT
MG813871.1 (Reference gene)	*Actin*	GTGCAACGCCTTCTCCGTTC	TGGCTGATGGAGCACAGAGG

**Table 2 molecules-27-05112-t002:** Effect of drought stress, bulk SA, and SANPs on plant height, shoot dry weight, root dry weight, and root–shoot ratio.

Soil Water Condition	Treatment	Plant Height (cm)	Shoot Dry Weight (gm)	Root Dry Weight (gm)	Root–Shoot Ratio
Normal irrigation	Control	57 ^c,d^ ± 0.33	4.07 ^c,d^ ± 0.18	4.14 ^b^ ± 0.21	1.07 ^b^ ± 0.067
	SANP (0.05 mM)	58 ^b,c^ ± 0.35	3.75 ^c,d^ ± 0.49	5.3 ^a^ ± 0.14	1.41 ^a^ ± 0.058
	SANP (0.1 mM)	61 ^a^ ± 2.3	5.7 ^a^ ± 0.66	4.3 ^b^± 0.17	0.76 ^d^ ± 0.01
	SA (0.05 mM)	56 ^c,d,e^ ± 0.18	4.63 ^a,b,c^ ± 0.43	2.94 ^c^ ± 0.15	0.63 ^e^ ± 0.012
	SA (0.1 mM)	60 ^a,b^ ± 2.31	5.63 ^a,b^ ± 0.35	4.42 ^b^ ± 0.20	0.78 ^d^ ± 0.015
Drought treatment	Control	48 ^i^ ± 0.17	1.05 ^f^ ± 0.24	0.15 ^d,e,f^ ± 0.22	0.12 ^g^ ± 0.009
	SANP (0.05 mM)	51 ^h^ ± 0.34	2.4 ^e^ ± 0.39	3.35 ^c^ ± 0.11	1.16 ^b^ ± 0.09
	SANP (0.1 mM)	55 ^e,f^ ± 1.15	1.3 ^e^± 0.34	1.08 ^d^ ± 0.27	0.83 ^c^ ± 0.012
	SA (0.05 mM)	53 ^f,g^ ± 0.44	1.12 ^f^ ± 0.10	0.60 ^d,e,f^ ± 0.18	0.13 ^f,g^ ± 0.015
	SA (0.1 mM)	53.5 ^e,f,g^ ± 0.88	1.4 ^e^ ± 0.61	0.84 ^d,e^ ± 0.31	0.60 ^e,f^ ± 0.017

Each value represents the mean ± SE of species in different treatments. Small letters represent significant differences among treatments tested using the LSD test. All tests were considered significant at *p* < 0.05.

**Table 3 molecules-27-05112-t003:** Effect of drought stress, bulk SA, and SANPs on chlorophyll a and b, carotenoid, and chlorophyll a + b (mg/gm of a plant leaf).

Soil Water Condition	Treatment	Chl-a (mg/gm)	Chl-b (mg/gm)	Carotenoids (mg/gm)	Chl-a + Chl-b (mg/gm)
normal irrigation	Control	0.95 ^b^ ± 0.006	0.35 ^b^ ± 0.0058	0.31 ^a^ ± 0.012	1.3 ^b^ ± 0.02
	SANP (0.05 mM)	0.99 ^a^ ± 0.0054	0.25 ^e^ ± 0.006	0.21 ^d^ ± 0.01	1.24 ^c^ ± 0.003
	SANP (0.1 mM)	0.94 ^b,c^ ± 0.012	0.23 ^f^ ± 0.009	0.31 ^a^ ± 0.015	1.17 ^de^ ± 0.014
	SA (0.05 mM)	0.91 ^d,e^ ± 0.015	0.22 ^f^ ± 0.015	0.32 ^a^ ± 0.003	1.03 ^g^ ± 0.012
	SA (0.1 mM)	0.92 ^cd^ ± 0.012	0.28 ^d^ ± 0.01	0.20 ^d^ ± 0.017	1.12 ^e,f^ ± 0.01
drought stress	Control	0.75 ^f^ ± 0.012	0.17 ^h^ ± 0.007	0.25 ^c^ ± 0.009	0.92 ^h^ ± 0.006
	SANP (0.05 mM)	0.96 ^b^ ± 0.015	0.16 ^a^ ± 0.12	0.29 ^b^ ± 0.009	1.52 ^a^ ± 0.044
	SANP (0.1 mM)	0.75 ^f^ ± 0.02	0.31 ^c^ ± 0.012	0.21 ^d^ ± 0.006	1.06 ^g^ ± 0.02
	SA (0.05 mM)	0.94 ^b,c^ ± 0.009	0.21 ^g^ ± 0.012	0.31 ^a^ ± 0.012	1.15 ^e,f^ ± 0.017
	SA (0.1 mM)	0.90 ^d,e^ ± 0.026	0.30 ^c^ ± 0.009	0.28 ^b^ ± 0.006	1.2 ^d^ ± 0.007

Each value represents the mean ± SE of species in different treatments. Small letters represent significant differences among treatments tested using the LSD test. All tests were considered significant at *p* < 0.05.

**Table 4 molecules-27-05112-t004:** Effect of drought stress, bulk SA, and SANPs on relative water content (RWC) and leaf area index (LAI).

Soil Water Condition	Treatments	Relative Water Content % (RWC)	Leaf Area Index
normal irrigation	Control	88.83 ^b^^,^^c^ ± 0.68	3.4 ^g,h^ ± 0.25
	SA-NPs (0.05 mM)	87.06 ^d,e^ ± 0.06	7.03 ^a^ ± 0.22
	SA-NPs (0.1 mM)	91.10 ^a^ ± 0.33	4.5 ^d,e^ ± 0.21
	SA (0.05 mM)	86.82 ^e,f^ ± 0.09	4.06 ^e,f^ ± 0.30
	SA (0.1 mM)	88.00 ^b,c,d^ ± 0.18	5.67 ^b^ ± 0.18
drought stress	Control	66 ^i^ ± 0.18	2.07 ^i^ ± 0.19
	SA-NPs (0.05 mM)	86.10 ^e,f^ ± 0.83	5.03 ^c^ ± 0.26
	SA-NPs (0.1 mM)	89.17 ^b^ ± 0.44	3.77 ^f,g^ ± 0.15
	SA (0.05 mM)	71.66 ^h^ ± 0.74	2.97 ^h^ ± 0.12
	SA (0.1 mM)	85.33 ^f^ ± 0.33	4.87 ^c,d^ ± 0.24

Each value represents the mean ± SE of species in different treatments. Small letters represent significant differences among treatments tested using the LSD test. All tests were considered significant at *p* < 0.05.

**Table 5 molecules-27-05112-t005:** Effect of drought stress, bulk SA, and SANP on total alkaloid percentage.

Soil Water Condition	Treatment	Total Alkaloid Percentage
Normal irrigation	Control	0.72 ^e^ ± 0.008
	SANP (0.05 mM)	0.75 ^d^ ± 0.012
	SANP (0.1 mM)	0.77 ^c^ ± 0.006
	SA (0.05 mM)	0.65 ^g^ ± 0.012
	SA (0.1 mM)	0.70 ^f^ ± 0.013
Drought stress	Control	0.77 ^c^ ± 0.006
	SANP (0.05 mM)	0.83 ^a^ ± 0.007
	SANP (0.1 mM)	0.80 ^b^ ± 0.007
	SA (0.05 mM)	0.76 ^c^ ± 0.009
	SA (0.1 mM)	0.82 ^a^ ± 0.0006

Each value represents the mean ± SE of species in different treatments. Small letters represent significant differences among treatments tested using the LSD test. All tests were considered significant at *p* < 0.05.

## Data Availability

The data presented in this study are available upon request from the corresponding author. The data are not publicly available due to privacy concerns.

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
