# Peer review of "Nanobiotechnological Approaches to Enhance Drought Tolerance in Catharanthus roseus Plants Using Salicylic Acid in Bulk and Nanoform"

_molecules, 2022, doi:10.3390/molecules27165112_

Round 1
Reviewer 1 Report
The Authors studied the effect of bulk salicylic acid and salicylic acid nanoparticles on water-stressed Catharanthus roseus plants. They stated that both SA and SA-NPs alleviated the negative effects of drought on treated plants by increasing shoot and root weights, relative water content, leaf area index, chlorophyll content and total alkaloids percentage.
Obtained results evidenced the importance of salicylic acid nanoparticles on induction of drought stress tolerance.
Although the manuscript is well conducted and well organized, and obtained results are interesting, in my opinion, it is not suitable for publication on Molecules. Indeed, no compounds are herein described, extracted, and characterized.
For example, Which alkaloids do they refer to? Why they don't extract and analyzed alkaloids?
In my opinion a different journal, i.e. Plants, would be more appropriate.
Author Response
Response to the reviewer (1)
The Authors studied the effect of bulk salicylic acid and salicylic acid nanoparticles on water-stressed Catharanthus roseus plants. They stated that both SA and SA-NPs alleviated the negative effects of drought on treated plants by increasing shoot and root weights, relative water content, leaf area index, chlorophyll content and total alkaloids percentage.
Obtained results evidenced the importance of salicylic acid nanoparticles in the induction of drought stress tolerance.
Although the manuscript is well conducted and well organized, and obtained results are interesting, in my opinion, it is not suitable for publication on Molecules. Indeed, no compounds are herein described, extracted, and characterized.
For example, Which alkaloids do they refer to? Why don’t they extract and analysed alkaloids?
In my opinion, a different journal, i.e. Plants, would be more appropriate.
Firstly, we would like to thank the reviewers for the valuable comments and constructive suggestions to our manuscript, which greatly helped to enhance the quality of our manuscript.
Secondly, the goal of this research is to use various SA concentrations in bulk and nano form to reduce the effects of drought stress on Catharanthus roseus plants. It also evaluates the effects of these concentrations on morphological, physiological, and biochemical traits, as well as stress-responsive gene expression in stressed plants.
Our goal did not target alkaloids or any other analysis for alkaloids, so we focused on our targeted goal.
We respect your comment, and we could run another research project targeting specific alkaloid compounds which may enhance resistance response to biotic or abiotic stressors.
Reviewer 2 Report
This manuscript reports important findings on the effects of bulk and nanoform SA in alleviating drought stress in Catharanthus roseus.
In general, the manuscript is well written. Having said that, I have a few comments that may further improve the quality of the manuscript. The comments can be found below and in the manuscript copy.
Introduction:
The second paragraph is rather long. I suggest splitting it into two. The introduction lacks information on the physiological and economic impact of drought on C. roseus. It can be added somewhere in the second paragraph.
Since the present introduction is already lengthy, I suggest combining the paragraphs on the role of genes in drought stress into one and making it brief.
The last paragraph/sentence of the introduction section is not very clear. Maybe the author can break it into two sentences.
Method:
2.2
The description of the method for the SA and SA-NP treatment is insufficient. How many treatments in total? How many pots of plants were used for each treatment? How did SA and SA-NP were applied? Were they applied one time or repeated throughout the 15 days?
Results
I suggest adding a title for the section that describes the characterization of SA nanoparticles (refer to the comment in the manuscript). In relation to that, the numbering for the rest of the result sections must be edited accordingly.
Line 218 - In this sentence, the author was not describing an 'increase' which should be calculated in relation to the control treatment and normally represented as a percentage. Instead, the author only reported the amount of dry weight obtained for shoot and root, as presented in Table 2. Therefore, the author must correct the writing style as mentioned here and in other places (highlighted in the manuscript) since it carries a different meaning.
Discussion
Too long. Can the author summarize or cut down some parts?

Author Response
"Authors’ response to reviewers’ comments"
Nanobiotechnological approaches to enhance drought tolerance in Catharanthus roseus plants using salicylic acid in the bulk and nanoform
Dina Salim1, Hoda A.S. El-Garhy2, Ismail A. Ismail3, Eldessoky S. Dessoky3, Bassem N. Samra3 and Tahsin Shoala4
Comments and replies
We would like to thank the reviewers for the valuable comments and constructive suggestions to our manuscript, which greatly helped to enhance the quality of our manuscript.
For the provided specific points:
we have made the requested modifications and hope the revised version would meet the requirements.
|
Reviewer 1 comments: |
Authors replies |
|
1- Which alkaloids do they refer to? Why they don't extract and analyzed alkaloids?
|
- C. roseus contain about one hundred and twenty alkaloids, seventy of which had pharmacological importance |
|
Reviewer 2 comments: |
|
|
1- The second paragraph is rather long. I suggest splitting it into two.
|
Done |
|
2- The introduction lacks information on the physiological and economic impact of drought on C. roseus. It can be added somewhere in the second paragraph. |
Done |
|
3- Since the present introduction is already lengthy, I suggest combining the paragraphs on the role of genes in drought stress into one and making it brief.
|
Done |
|
4- The last paragraph/sentence of the introduction section is not very clear. Maybe the author can break it into two sentences. |
Done |
|
5- The description of the method for the SA and SA-NP treatment is insufficient. How many treatments in total? How many pots of plants were used for each treatment? How did SA and SA-NP were applied? Were they applied one time or repeated throughout the 15 days? |
The requested information were added to M&M section |
|
6- I suggest adding a title for the section that describes the characterization of SA nanoparticles (refer to the comment in the manuscript). In relation to that, the numbering for the rest of the result sections must be edited accordingly.
|
Done |
|
7- Line 218 - In this sentence, the author was not describing an 'increase' which should be calculated in relation to the control treatment and normally represented as a percentage. Instead, the author only reported the amount of dry weight obtained for shoot and root, as presented in Table 2. Therefore, the author must correct the writing style as mentioned here and in other places (highlighted in the manuscript) since it carries a different meaning. 8- Discussion Too long. Can the author summarize or cut down some parts? |
The writing style has been corrected in all the ms as requested
Done |
|
Reviewer 3 comments: |
|
|
1- Line 61. Please avoid to start the sentence with abbreviations. Please check throughout the manuscript. |
Done |
|
2- Materials and methods 2.3-2.5 how much old plants were used in case of SFW, RFW, LAI, RWC, and pigments measurements? |
(60 days old), mentioned in the required sections |
|
3- In 3.1 please specify SA and SA-NPs how much increase and decrease the plant growth responses. For example, SA application increased plant height X% compared to control under stress conditions. OR drought stress how much reduces growth. Similarly, improve 3.2 section. |
Done |
|
4- Line 240-41; RWC by 71.66 and 85.33, use %. |
Done |
|
5- Line 316; Authors should give more relevant hypothesis on growth reduction by drought. They did not check stomatal movement, so, this is not proper statement. |
Done |
|
6- Recent literature are missing. Discussion can improve comparing with recent refences. |
Recent litratures were used |
|
7- The conclusion should be more precise and brief. |
Done |
|
8- Manuscripts contain many grammatical and typo mistakes. |
Checked well |

Reviewer 3 Report
The authors target a good topic and did nice work. Introduction, methodology, and results are very well-presented. The manuscript can be accepted. However, the following points need to consider.
Line 61. Please avoid to start the sentence with abbreviations. Please check throughout the manuscript.
Materials and methods 2.3-2.5 how much old plants were used in case of SFW, RFW, LAI, RWC, and pigments measurements?
In 3.1 please specify SA and SA-NPs how much increase and decrease the plant growth responses. For example, SA application increased plant height X% compared to control under stress conditions. OR drought stress how much reduces growth. Similarly, improve 3.2 section.
Line 240-41; RWC by 71.66 and 85.33, use %.
Line 316; Authors should give more relevant hypothesis on growth reduction by drought. They did not check stomatal movement, so, this is not proper statement.
Recent literature are missing. Discussion can improve comparing with recent refences.
Conclusion should be more precise and brief.
Manuscripts contain many grammatical and typo mistakes.
Author Response

(The authors gave the same response as above.)

Round 2
Reviewer 1 Report
The manuscript has been properly improved